# Genome-Wide Identification and Expression Analysis of Calmodulin and Calmodulin-like Genes, Revealing *CaM3* and *CML13* Participating in Drought Stress in *Phoebe bournei*

**DOI:** 10.3390/ijms25010545

**Published:** 2023-12-30

**Authors:** Ningning Fu, Li Wang, Xiao Han, Qi Yang, Yuting Zhang, Zaikang Tong, Junhong Zhang

**Affiliations:** State Key Laboratory of Subtropical Silviculture, School of Forestry & Biotechnology, Zhejiang A&F University, Lin’an, Hangzhou 311300, China; paodiule@stu.zafu.edu.cn (N.F.); 2020602041101@stu.zafu.edu.cn (L.W.); hanx2017@zafu.edu.cn (X.H.); qiyang@zafu.edu.cn (Q.Y.); zhangyt@zafu.edu.cn (Y.Z.)

**Keywords:** *Phoebe bournei*, CaM/CML gene family, *PbCaM3*/*CML13*, drought stress

## Abstract

Calmodulin (CaM) and calmodulin-like (CML) proteins are major Ca^2+^ sensors involved in the regulation of plant development and stress responses by converting Ca^2+^ signals into appropriate cellular responses. However, characterization and expression analyses of *CaM*/*CML* genes in the precious species, *Phoebe bournei*, remain limited. In this study, five *PbCaM* and sixty *PbCML* genes were identified that only had EF-hand motifs with no other functional domains. The phylogenetic tree was clustered into 11 subgroups, including a unique clade of *PbCaM*s. The *PbCaM*s were intron-rich with four EF-hand motifs, whereas *PbCML*s had two to four EF-hands and were mostly intronless. *PbCaM*s/*CML*s were unevenly distributed across the 12 chromosomes of *P. bournei* and underwent purifying selection. Fragment duplication was the main driving force for the evolution of the PbCaM/CML gene family. Cis-acting element analysis indicated that *PbCaM*s/*CML*s might be related to hormones, growth and development, and stress response. Expression analysis showed that *PbCaM*s were generally highly expressed in five different tissues and under drought stress, whereas *PbCML*s showed specific expression patterns. The expression levels of 11 candidate *PbCaM*s/*CML*s were responsive to ABA and MeJA, suggesting that these genes might act through multiple signaling pathways. The overexpression of *PbCaM3*/*CML13* genes significantly increased the tolerance of yeast cells to drought stress. The identification and characterization of the CaM/CML gene family in *P. bournei* laid the foundation for future functional studies of these genes.

## 1. Introduction

Water is an important ecological factor that influences the growth and development of plants throughout their life. The vast majority of metabolic activities in plants must be mediated by water. For instance, the absorption and transportation of nutrients, as well as photosynthesis, respiration, transpiration, and other physiological actions of plants, are inseparable from the participation of water [1]. Drought generally causes widespread and negative effects on plants, such as slow cell division and elongation, reduced photosynthesis and transpiration, increased ROS, and unbalanced osmotic pressure [2,3]. Drought stress is one of the major environmental factors influencing the geographical distribution and biomass accumulation of trees [4]. The frequency and intensity of drought will continue to increase in the future [5].

Understanding the mechanisms by which plants sense and transmit drought stress signals to initiate adaptive responses is important for improving plant drought tolerance. Plants respond to environmental changes by activating signaling cascades to control and synergize physiological and biochemical reactions [6]. Ca^2+^, as a major intracellular second signaling molecule, is involved in many signaling pathways, including internal signaling and external stimulus responses [7]. There is growing evidence that a variety of external stresses, such as light, salinity, heat, cold, drought, and mechanical damage, can rapidly cause an increase in intracellular Ca^2+^ concentrations [8]. Associated changes in intracellular Ca^2+^ concentration are manifested as calcium signaling, and the conformational changes induced by Ca^2+^-binding proteins in response to Ca^2+^ can further modulate the activity of downstream targets, thereby delivering Ca^2+^ signaling [9].

Calcium-binding protein sensors can be classified into four groups: calmodulins (CaMs), CaM-like proteins (CMLs), Ca^2+^-dependent protein kinases (CDPKs), and calcineurin B-like proteins (CBLs) [10]. In most Ca^2+^ sensors, the EF-hand (a helix–loop–helix structure) is the most common pattern identified as the Ca^2+^-binding site [11]. CaM/CML proteins do not have any functional domains other than the EF-hand, unlike CDPK and CBL proteins. CaMs are highly conserved Ca^2+^-binding proteins in eukaryotes that typically contain four EF-hand motifs [12]. Calmodulin-like proteins (CMLs) belong to plant-specific Ca^2+^ sensors. Many genes encoding CML proteins have been identified as containing 1–6 EF-hands with no other known functional structural domains, and they share 30.2–84.6% sequence homology with CaMs. The identification of the CML gene family has been achieved in several plants, such as *Arabidopsis thaliana*, *Zea mays*, and *Solanum lycopersicum* [13,14,15]. These studies have shown that *CML* members vary widely in the sequence, length, and number of EF-hand motifs. Furthermore, in the genome-wide characterization of the CaM/CML gene family, there are more genes encoding CML proteins than CaM proteins in plant species [16]. For example, there are 168 and 54 *CML* members in *Brassica napus* and *Phalaenopsis equestris*, respectively, but only 25 and 4 corresponding *CaM* members [17,18].

As important Ca^2+^ sensors in plants, *CaM*s and *CML*s have important complexities in the signaling networks that affect plant growth, development, and response to environmental changes [19]. For example, the *cml24* mutant significantly reduced the rate of pollen germination and pollen tube growth [20]. *CaM*s and *CML*s are also involved in abiotic stresses in plants, such as salt stress, drought stress, and cold stress. In tomato, *SlCML37* transgenic tomato fruits significantly improved tolerance to low-temperature stress and may play a role in regulating the plant low-temperature response by interacting with *SlUMP1*, affecting proteasome assembly activity and modulating the degradation of target proteins [21]. In *Medicago truncatula*, salt stress, osmotic stress, and ABA treatment up-regulated *MtCML40* expression, and plants overexpressing *MtCML40* showed severe growth inhibition and reduced the photosynthetic rate under NaCl compared with WT, suggesting that *MtCML40* negatively regulates salt tolerance [22]. The overexpression of *OsDSR-1*, a *CML* gene, in transgenic plants caused higher ABA sensitivity, ROS enzyme activity, and transcript levels in multiple ROS scavenging and stress-related genes, and these were more drought tolerant under drought stress compared with WT and *OsDSR-1*-Ri plants [23]. Moreover, ABA is the primary signal for plants to cope with drought and plays a crucial role in various physiological processes of their life cycle, such as stomatal closure, root regulation, the transcriptional activation of related genes, and metabolic alterations. Under drought conditions, endogenous ABA levels in plants are usually elevated to enhance drought tolerance [24]. *AtCML9* expression was rapidly induced by ABA in *Arabidopsis* seedlings, and seed germination and seedling growth of the *cml9* knockout mutant showed hypersensitivity to ABA [25].

*Phoebe bournei* is a subtropical evergreen broad-leaved tree species, preferring humidity and shade, with a long lifespan and slow growth, and seed reproduction is the main way of its natural renewal. Its wood is known as “noble wood” because of its strong resistance to decay, special fragrance, and unique golden–yellow texture, and it is also an important afforestation and ornamental tree species in the mountainous areas of southern China [26,27]. *P. bournei* is mainly found in areas with a warm and humid climate and abundant rainfall, such as Jiangxi, Fujian, Zhejiang, Guangdong, Guangxi, Guizhou, Hubei, and Hunan provinces [28]. However, in recent years, over-harvesting and habitat degradation have led to a reduction in the size and fragmentation of the natural population of *P. bournei*, seriously threatening its survival. At present, a number of protected areas have been established within its range and *P. bournei* is being propagated and promoted as a silvicultural species [29]. Therefore, the study of the response mechanism of *P. bournei* to drought stress is very important for the cultivation and conservation of this species. As important Ca^2+^ sensors, *CaM*s and *CML*s are widely involved in plant responses to abiotic stresses and have been genome-wide analyzed in various model plants and crops [30,31]. However, the CaM/CML gene family of *P. bournei* have not been systematically identified and characterized. In this study, based on our previous genome and transcriptomic data [26], 5 *CaM* genes and 60 *CML* genes were identified from the genome of *P. bournei* and their protein physicochemical properties, phylogenetic relationships, gene structure and conserved motifs, chromosomal localization and homology, and cis-acting elements in the promoter regions were characterized. The expression profiles of *PbCaM*s/*CML*s in five tissues and PEG treatment were determined, as well as their responses to ABA and MeJA treatments. The heterologous overexpression of candidate genes *PbCaM3* and *PbCML13* in INVSc1 yeast significantly increased the tolerance of yeast cells to drought stress. This study lays a foundation for the further investigation of *PbCaM*s/*CML*s functions and provides theoretical support for molecular-assisted breeding to produce new drought-resistant germplasm of *P. bournei*.

## 2. Results

### 2.1. Identification and Characterization of CaM/CML Genes in P. bournei

Based on the *Arabidopsis CaM*s/*CML*s and EF-hand motif sequences (PF00036), a total of 5 *PbCaM* and 60 *PbCML* members were identified in *P. bournei*. The corresponding protein sequences were further analyzed through NCBI CDD and SMART to confirm the presence of the EF-hand motif with no other functional domains. Based on their chromosomal localization, all *PbCaM*/*CML* genes were sequentially named from *PbCaM1* to *PbCaM5* and *PbCML1* to *PbCML60*, respectively (Appendix A).

All of the PbCaMs had four EF-hand domains and PbCMLs contained two to four EF-hands. The amino acid length of PbCaM/CML proteins ranged from 84 aa (PbCML8/26/34) to 442 aa (PbCML18), with the molecular weights ranging from 9.21 kDa (PbCML8) to 48.60 kDa (PbCML18), and the isoelectric points ranged from 4.04 (PbCML13/29) to 6.97 (PbCML47). The instability coefficients of these proteins ranged from 20.06 (PbCML8) to 68.51 (PbCML39), and the overall mean of the hydrophilicity scores (GRAVY) were all negative. Subcellular localization predictions indicated that most of the PbCaMs/CMLs were nuclear, cytoplasmic, and chloroplast proteins (Appendix A).

### 2.2. Phylogenetic Analysis of CaM/CML Genes in P. bournei

To gain insights into the evolutionary relationships and potential functions of *PbCaM*s/*CML*s, the phylogenetic tree was constructed with AtCaMs/CMLs as the reference proteins. PbCaM/CML proteins were classified into 11 subfamilies, each containing a different number of members, with the PbCaMs in a separate clade from the PbCMLs (Figure 1). Subfamily 8 had the largest number of members (13), followed by subfamilies 7 and 5 (10). Subfamily 9 was the smallest, containing only two members (PbCML4/24) (Appendix A). The CaM/CML proteins from *P. bournei* and *Arabidopsis* shared high degrees of similarity, suggesting similar functions among the homologous members.

### 2.3. Gene Structure and Conserved Motifs Analysis of PbCaMs/CMLs

The gene structure and conserved motifs of the PbCaM/CML gene family were explored based on the phylogenetic relationships. Conserved motif analysis revealed that motifs 1 and 2 were present in all PbCaMs/CMLs, which were functionally conserved motifs. All PbCaM proteins contained the typical four EF-hands. Of the 60 PbCMLs, nearly half (29) contained four EF-hands, thirteen contained three EF-hands, and the remaining eighteen only contained two EF-hands. The motif composition of PbCaMs/CMLs was shown to correspond to their phylogenetic relationships. For example, PbCaM/CML members in subfamily 11 contained two conserved motifs, whereas subfamilies 1, 2, 4, 9, and 10 contained four conserved motifs (Figure 2B). There were eight different types of EF-hand domains, and gene members of the same subfamily had similar domain types (Figure 2C). The analysis of the intron–exon structure of *PbCaM*s/*CML*s showed that the number of exons ranged from one to nine. Most *PbCaM*s/*CML*s contained only one exon. PbCML2/56 had the highest number of exons (nine), followed by PbCML41/58 (eight) (Figure 2D). *PbCaM*s/*CML*s with multiple exons mainly belonged to subfamilies 5 and 11, suggesting that the evolutionarily closely related *PbCaM*s/*CML*s have similar gene structures.

### 2.4. Cis-Regulatory Elements Analysis in the Promoters of PbCaM/CML Genes

Cis-regulatory elements (CREs) play a vital role in gene expression. Various hormone response elements, growth and development elements, and stress response elements were identified in the promoter regions of *PbCaM*/*CML* genes (Figure 3A–C, Appendix A). Hormone responses included abscisic acid (156), gibberellin (72), auxin (43), MeJA (174), and salicylic acid (48). The growth and development elements were light response elements (691), circadian control elements (14), seed-specific regulation elements (8), meristem expression elements (35), and endosperm expression elements (16). Stress response elements mainly included anaerobic induction (148), low-temperature response elements (52), defense and stress response elements (38), (MYB) drought-induced elements (66), and (MYB) flavonoid biosynthetic genes regulation elements (4). Almost all *PbCaM*s/*CML*s contained ABA response elements (ABRE) in the promoter regions, with *PbCML53* having the highest number of 11, followed by *PbCML19* (8). Most members contained MeJA response elements, with *PbCML9* in subfamily 8 and *PbCML2* in subfamily 5 containing 12 and 10, respectively. In addition, 37 members contained drought-induced response elements, with *PbCML16* from subfamily 8 containing the most (7). The flavonoid synthesis gene regulatory elements were found only in *PbCML1*/*5*/*23*/*36* from subfamilies 10/11/11/7, respectively.

### 2.5. Chromosome Locations and Synteny Analysis of PbCaM/CML Genes

The 60 *PbCML*s were unevenly distributed across all 12 chromosomes of *P. bournei* (Figure 4A). For example, chromosomes 1 and 2 contained ten *CML* genes, whereas chromosomes 6, 7, and 11 each contained only two *CML*s. Five *PbCaM* genes were located on chromosomes 1, 2, and 3, respectively. The analysis of gene duplication events in *PbCaM*s/*CML*s showed that 42 members were involved in 32 segmental duplications and 9 genes were involved in 5 tandem duplications. All 37 gene pairs involved in gene duplication had Ka/Ks ratios less than 1 and may have undergone purifying selection during evolution (Appendix A).

To explore the evolution of *PbCaM*s/*CML*s and their affinities with different species, the collinearity of *CaM*s/*CML*s between *P. bournei* and *A. thaliana*, *O. sativa*, *P. trichocarpa*, and *C. kanehirae* was investigated (Figure 4B). A total of 60 *PbCaM*s/*CML*s were collinearly associated with *C. kanehirae*, followed by *P. trichocarpa* (49), *O. sativa* (24), and *A. thaliana* (22) (Appendix A). Thirteen *PbCML*s (*PbCML1*/*3*/*12*/*20*/*32*/*33*/*38*/*41*/*49*/*51*/*55*/*56*/*60*) were collinearly associated among all four species.

### 2.6. Expression Patterns of PbCaMs/CMLs in Different Tissues

In order to detect the spatio-temporal expression patterns of *PbCaM*/*CML* genes, the expression profiles of 65 *PbCaM*s/*CML*s in five tissues of *P. bournei*, including leaf, stem bark, stem xylem, root bark, and root xylem, were characterized using the FPKM values (Figure 5A, Appendix A). After filtering the low-expressed genes, 47 *PbCaM*s/*CML*s were retained. Most members from the same cluster showed similar expression patterns, with some genes preferentially expressed in specific tissues. For example, *PbCML48*/*39*/*56* from subfamily 5 and *PbCML13* from subfamily 2 were highly expressed in leaves; *PbCML15*/*23*/*12* were preferentially expressed in root bark; *PbCML3*/*17*/*25*/*55* and *PbCML27*/*22*/*28*/*51* were preferentially expressed in the xylem of the stem and root, respectively. Five *PbCaM* genes (*PbCaM1*/*2*/*3*/*4*/*5*) were highly expressed in five tissues.

The expression profiles of 11 *PbCaM*s/*CML*s in five tissues were analyzed using qRT-PCR (Figure 5B). In agreement with the transcriptome data, *PbCML56* was highly expressed in leaves, which was 8.9 times higher than that in root xylem. *PbCaM3* and *PbCML30*/*31*/*42* were highly expressed in stem bark, with *PbCaM3* being the highest. *PbCML13* and *PbCML5* had the highest expression level in root bark. *PbCaM5* was specifically highly expressed in stem xylem, with its expression being 18 times higher than that in leaves. *PbCML57* was highly expressed in both stem and root bark, and *PbCML40*/*58* was highly expressed in both stem and root xylem.

### 2.7. Expression Patterns of PbCaM/CML Genes in Response to Drought Stress

To investigate the potential functions of *PbCaM*s/*CML*s in response to abiotic stress, the expression patterns of *PbCaM*s/*CML*s under PEG treatment were analyzed by available FPKM data (Figure 6A, Appendix A). A total of 46 *PbCaM*s/*CML*s were preserved after the removal of low expressed genes. The expression levels of *PbCML23*/*51* in subfamilies 11 and 7 were highest in CK and down-regulated with PEG treatment. The expression levels of *PbCML9*/*12*/*22*/*24*/*27*/*44* were significantly increased after 1 h of PEG treatment, and then decreased, and *PbCML15*/*35*/*39*/*45*/*55* were highly expressed after 1 d of PEG treatment. However, the expression levels of *PbCaM3*/*5* and *PbCML5*/*13*/*30*/*31*/*42* peaked after 3 d of PEG treatment.

The expression patterns of 11 genes induced by PEG were also analyzed using qRT-PCR (Figure 6B). Consistent with the transcriptome data, all the genes were induced by PEG with different fold changes. The expression level of *PbCML56* peaked after 1 d of PEG treatment, while *PbCaM3* had the highest expression after both 1 d and 3 d of PEG treatment with the highest FPKM value. The remaining genes were highly expressed after 3 d of PEG treatment, with *PbCaM5* and *PbCML57* being the most up-regulated genes, 5.7-fold and 5.3-fold higher than the control, respectively.

### 2.8. Expression Patterns of PbCaM/CML Genes in Response to ABA Treatment

The expression patterns of 11 *PbCaM*s/*CML*s under ABA treatment were performed by qRT-PCR (Figure 7). The expression levels of 11 genes were increased under ABA treatment with varying fold changes. *PbCML57* had the highest expression level after 3 h of ABA treatment, and *PbCML13* was highly expressed at both 3 and 12 h. *PbCML30*/*31*/*40*/*56*/*58* were most abundantly expressed after 24 h of ABA treatment. The expression of *PbCaM3* was maintained at a high level after 12, 24, and 48 h of ABA treatment.

### 2.9. Expression Patterns of PbCaM/CML Genes in Response to MeJA Treatment

The expression patterns of 11 *PbCaM*s/*CML*s under MeJA treatment were also performed using qRT-PCR (Figure 8). These genes were differentially induced by MeJA. *PbCaM3* and *PbCML13*/*57* had the highest expression level after 24 h of MeJA treatment, while *PbCML31*/*40*/*56*/*58* had the highest expression level after 48 h of MeJA treatment. *PbCaM5* and *PbCML5*/*30*/*42* were highly expressed after both 24 and 48 h of MeJA treatment. In total, 39 members of the 65 *PbCaM*s/*CML*s contained both ABA (ABRE) and MeJA (CGTCA-motif/TGACG-motif) response elements. It suggests that most *PbCaM*/*CML* genes may regulate plant response to drought stress by participating in multiple signal transduction pathways.

### 2.10. PbCaM3/PbCML13 Increased the Drought Tolerance of Yeast Cells

Drought response elements and stress response elements (TC-rich repeats) were abundantly present in the *PbCaM3*/*PbCML13* promoters. The expression levels of *PbCaM3*/*PbCML13* were increased under PEG, ABA, and MeJA treatments. To verify the function of *PbCaM3*/*PbCML13*, pYES2-*PbCaM3*/*PbCML13* recombinant plasmids were constructed and transformed into INVSc1 yeast receptor cells. After confirming that *PbCaM3*/*CML13* had successfully transformed yeast cells, yeast drought stress tolerance experiments were performed. Under normal conditions, yeast containing pYES2-*PbCaM3*/*PbCML13* showed no significant differences in growth size and status compared with the empty vector (pYES2) (Figure 9A). However, the yeast with *PbCaM3*/*PbCML13* grew better than the control under 5 mM PEG conditions. In particular, the number of yeast clones with *PbCaM3*/*PbCML13* were significantly higher than that of the control after 10^−3^-fold and 10^−4^-fold dilution (Figure 9B). *PbCaM3*/*PbCML13* heterologous transformed yeast improved the tolerance of cells to drought stress.

## 3. Discussion

Calcium is not only an essential nutrient for plant growth and development but also a crucial signaling molecule that facilitates a wide array of physiological processes [32]. Ca^2+^ sensor proteins are key proteins in the control of intracellular Ca^2+^ homeostasis by binding to Ca^2+^ and modulating downstream targets in response to a variety of stimuli-induced Ca^2+^ fluctuations and signaling transduction [33,34]. CaM proteins and CML proteins, as crucial members of Ca^2+^ sensors, are highly conserved in eukaryotes and specific in plants, respectively. It is noteworthy that these proteins solely possess the EF-hand domain, lacking any other functional domains [35]. The identification of *CaM*/*CML* genes has been completed in a number of model plant species, such as *Arabidopsis* [13], rice [36], tomato [15], and others [37,38]. However, the identification and functional analysis of the CaM/CML gene family in non-model species have not been fully investigated. *P. bournei*, as a source of “golden Nanmu” wood, has high economic and ecological values and is widely afforested in the mountainous areas of southern China. In this study, five *PbCaM* and sixty *PbCML* genes were identified in the whole genome of *P. bournei*, which showed extensive differences in protein length, molecular weight, and theoretical isoelectric point (Appendix A), indicating the diversity of *PbCaM*s/*CML*s.

The number of *PbCaM*s/*CML*s was similar to that of grapevine (three *VviCaM*s and sixty-two *VviCML*s) [39], more than that of *Arabidopsis* (six *AtCaM*s and fifty *AtCML*s) [13] and apple (four *MdCaM*s and fifty-eight *MdCML*s) [40], but less than that of *B. napus* (25 *BnaCaM*s and 168 *BnaCML*s) [17], which suggests that the evolutionary process of *PbCaM*s/*CML*s is not identical to that of other species. The predicted number of EF-hands were four in *PbCaM*s and two–four in *PbCML*s, which is consistent with those in apple [40]. Based on phylogenetic relationships, PbCaM/CML proteins were clustered into 11 subgroups (Figure 1), which was the same as that of *B. napus* [17], less than *Nelumbo nucifera* (twelve subgroups) [41], and more than wheat (seven subgroups) [38]. All *PbCaM* genes were clustered in a unique subgroup. The clustering of PbCaM/CML proteins with AtCaMs/CMLs showed their evolutionary relationships and potential functional similarities, providing a useful reference for exploring the gene functions of *PbCaM*s/*CML*s [40]. It has been reported that in plants, most *CML*s are intronless, whereas *CaM*s are intron-rich [42]. Partial duplications have probably affected the intron numbers in a gene. In addition, the number of introns/exons can alter the speed of the gene expression process and the mRNA processing, suggesting that genes with a low number of introns/exons may be faster expressed [43,44]. Similarly, in our study, *PbCaM*s in subgroup 1 and *PbCML*s in subgroups 2/5/11 were intron-rich, whereas other *PbCML*s, such as those in subgroups 6 and 9, were intronless (Figure 2D). The results suggest that *CaM*/*CML* genes are structurally diverse and conserved in different plant species, which may lead to functional diversity.

Chromosomal localization results showed that *PbCaM*s/*CML*s were unevenly distributed on all 12 chromosomes of *P. bournei*, suggesting that the evolution of *PbCaM*s/*CML*s was shaped by segmental duplication. Gene duplication analysis further confirmed that segmental duplications (32) were much larger than tandem duplications (5) as the main driving force for the evolution of *PbCaM*s/*CML*s (Figure 4A). All gene pairs involved in gene duplications were classified into the same subgroup, and the Ka/Ks ratios were less than 1, suggesting that *PbCaM*/*CML* genes are highly conserved in the process of family expansion (Appendix A). The specific expression of genes is closely linked to cis-acting elements in their promoter regions. Previous studies have shown that certain *CaM*/*CML* genes with specific cis-acting elements are involved in hormonal or abiotic stress responses [45]. For example, the promoter region of the *AtCML9* gene (homologue of *PbCML13*) was enriched with ABRE and GT1-box elements, and its expression was significantly induced by salinity, drought, and ABA treatments [25]. In *P. bournei*, *PbCaM*s/*CML*s contained various cis-elements related to hormones, growth and development, and stress (Figure 3). More than half of the *PbCaM*s/*CML*s contained cis-acting elements associated with ABA (ABRE), MeJA (CGTCA-motif/TGACG-motif), and drought stress (MBS), suggesting that *PbCaM*s/*CML*s may be extensively involved in the ABA and JA pathways to regulate drought.

The significant functions of *CaM*s/*CML*s in plant development and stress tolerance have been extensively reported. For instance, transgenic *Arabidopsis* seedlings overexpressing *CAM7* exhibited a stronger inhibition of hypocotyl elongation and higher chlorophyll content under light at multiple wavelengths [46]. *AtCML39* is involved in the regulation of seed germination and fruit development in *Arabidopsis*, and *CML39* deletion results in shorter siliques and reduced seed as well as ovule numbers [47]. In addition, *CaM*s/*CML*s are also involved in the response to various abiotic stresses such as drought and salinity. In *Solanum pennellii*, *SpCaM6* transcription in stems and roots was highly induced by drought, salt, and ABA treatments [48]. In rice, *OsCML4* confers drought tolerance by scavenging ROS in plants and inducing the expression of stress-related genes [49]. We determined the expression patterns of *PbCaM*s/*CML*s in different *P. bournei* tissues and under drought stress using RNA-seq and qRT-PCR analyses to assess their potential functions (Figure 5 and Figure 6). It was found that the evolutionarily conserved *CaM*s exhibited a broad spatiotemporal expression profile in five different tissues, whereas most of the *CML*s showed more tissue-specific expression patterns. Among all *PbCaM*s, *PbCaM3* had high expression levels in all five tissues, suggesting that *PbCaM3* has an important and pervasive function in *P. bournei* trunk formation and overall growth and development. In addition, some *PbCML*s were highly expressed in specific tissues, such as *PbCML48* in leaves, *PbCML23* in stem bark and root bark, *PbCML44* in stem xylem, and *PbCML27* in root xylem. Similarly, the expression of *PbCaM*s was maintained at a high level under drought treatment, with *PbCaM2*/*3* being the highest, whereas *PbCML*s were only expressed at specific times and stages, for example, *PbCML27*/*55*/*13* were highly expressed at 1 h, 1 d, and 3 d, respectively.

It has been shown that ABA and MeJA are the two main signals used by plants to cope with drought [50,51]. The expression patterns of *PbCaM*s/*CML*s in response to ABA and MeJA may be helpful in exploring the functional members of stress tolerance. Eleven selected *PbCaM*s/*CML*s were analyzed using qRT-PCR in response to ABA and MeJA (Figure 7 and Figure 8). All of these genes were differentially induced by ABA and MeJA to up-regulate their expression. Most members were expressed at higher levels during the mid-treatment period (24–48 h), with the expression of *PbCML13* increasing more than 14-fold after 12 h of ABA treatment, and the expression of *PbCaM5* increasing nearly 9-fold after 48 h of MeJA treatment. We screened two drought tolerance candidate genes, *PbCaM3* and *PbCML13*, and when they were overexpressed into INVSc1 yeast, the transgenic yeast cells grew significantly better under drought conditions compared with the control (Figure 9), suggesting that *PbCaM3* and *PbCML13* may play a role in the drought tolerance of *P. bournei*, consistent with the results of Jia et al. [52]. In summary, the identification and expression analysis of *PbCaM*s/*CML*s lays the foundation for future functional studies and is important for advancing the molecular breeding of drought tolerance in *P. bournei*.

## 4. Materials and Methods

### 4.1. Identification and Analysis of the CaM/CML Gene Family in P. bournei

The AtCaM/CML protein sequences retrieved from the TAIR database (https://www.arabidopsis.org/, accessed on 22 October 2023) [53] were used to perform a BLASTp homology search against *P. bournei* proteins with an e-value of 10^−5^ (Appendix A). Moreover, a hidden Markov model (HMM) profile of EF-hand motif (PF00036) was obtained from the Pfam database [54] and used to search the *P. bournei* local protein database using the HMM search tool (Appendix A) [55]. After removing redundancy from the BLAST and HMMER hits, the remaining sequences were uploaded to SMART [56] and the NCBI Conserved Domain Database [57] to confirm the existence and integrity of the EF-hand domain without any other domains. The identified *PbCaM*/*CML* genes were named according to their location on the *P. bournei* chromosomes. The physicochemical properties of the PbCaM/CML proteins were analyzed using the ExPASy ProtParam tool [58] and subcellular localization was predicted by the WOLF PSORT online website [59].

### 4.2. Sequence Alignment and Phylogenetic Analysis of PbCaM/CML Proteins

The multiple sequence alignment of PbCaM/CML and AtCaM/CML proteins was performed using the ClustalW tool [60]. Using the neighbor-joining (NJ) method, a phylogenetic tree was constructed through the MEGA-X software, with a bootstrap of 1000 [61]. The output evolutionary tree was annotated and embellished in groups using the iTOL online website (https://itol.embl.de/, accessed on 27 October 2023).

### 4.3. Gene Structure and Conserved Motifs Analysis of PbCaMs/CMLs

The conserved motifs of PbCaM/CML proteins were predicted via the MEME online website, setting the number of motifs to 4 [62]. Based on the *P. bournei* genome annotation file, a joint analysis of the intron–exon structure and conserved motifs of *PbCaM*s/*CML*s based on phylogenetic relationships was performed using TBtools software [63].

### 4.4. Analysis of Cis-Acting Elements in the Promoters of PbCaMs/CMLs

Based on the full-length of *P. bournei* DNA sequences, 2000 bp sequences upstream of the *PbCaM*s/*CML*s transcription start site were extracted using TBtools and submitted to the PlantCARE database to predict their cis-regulatory elements [64].

### 4.5. Chromosomal Locations and Synteny Analysis of PbCaMs/CMLs

*A. thaliana*, *Oryza sativa*, *Cinnamomum kanehirae*, and *Populus trichocarpa* genome data were downloaded from the NCBl database (https://www.ncbi.nlm.nih.gov/, accessed on 3 November 2023). Using genome files and gene annotation files for the corresponding species, association analyses were carried out within the *P. bournei* genome and between *P. bournei* and other species using the MCScanX tool in TBtools. Visualization of gene complexes and anchoring of *PbCaM*/*CML* genes to the corresponding chromosomes were conducted using Advanced Circos and Dual Synteny Plot tools. The Ka/Ks ratios of homologous gene pairs involved in the *P. bournei* gene duplication were calculated using KaKsCalculator (3.0). The evolutionary divergence time of the PbCaM/CML gene family was calculated according to the equation T = Ks/2λ (where λ = 1.5 × 10^−8^ in dicotyledons) [65].

### 4.6. Expression Patterns of PbCaMs/CMLs in Different Tissues

The five different tissues (including leaf, stem bark, root bark, stem xylem, and root xylem) and transcriptome data used for the analysis of spatiotemporal expression patterns of *PbCaM*/*CML* genes were obtained from Song et al. [66]. Heatmaps were constructed using the Lianchuan Cloud platform (https://www.omicstudio.cn/analysis, accessed on 15 November 2023) [67].

### 4.7. Plant Materials and Treatments

Plant materials were selected from 1.5-year-old cultivars of *P. bournei* ‘Wuyuan 8’ with uniform growth. The experiment was conducted with four replicates of six seedlings each. For the drought treatment, container seedlings were debagged and pre-cultured in 1/4 Hoagland nutrient solution for 3 months. Subsequently, treated groups were induced with 10% PEG and untreated plants served as control and sampled at 1, 24, and 72 h. The collected samples were snap-frozen in liquid nitrogen and stored at −80 °C. The expression patterns of *PbCaM*/*CML* genes under drought stress were analyzed based on the well-established transcriptome database by our team.

For hormone treatments, plants were sprayed evenly with 1 mM ABA solution until the soil surface was completely moistened, and samples were taken at 1, 3, 12, 24, and 48 h using sprayed water as control. Seedlings were treated with 2 mM MeJA in the same way, and untreated plants were used as control and sampled at 0, 3, 12, 24, 48, and 96 h. The samples were immediately processed as described above.

### 4.8. RNA Isolation and Gene Expression Analysis

Total RNA was extracted using the M5 Plant RNeasy Complex Mini Kit (MF045, Mei5bio, Beijing, China) and reverse transcription was performed using the HiScript III All-in-one RT SuperMix Perfect for qPCR (R333, Vazyme, Nanjing, China). Quantitative real-time PCR (qRT-PCR) was carried out using the ChamQ SYBR Color qPCR Master Mix (Q411, Vazyme, Nanjing, China) on a CFX-96-well real-time PCR system (Bio-Rad, Hercules, CA, USA). The qRT-PCR reaction procedure was a two-step amplification method: pre-denaturation phase 95 °C for 30 s; cyclic reaction phase 95 °C for 10 s and 60 °C for 30 s, repeated for 40 cycles; melting phase 95 °C for 15 s, 60 °C for 60 s, and 95 °C for 15 s. The fold change in target genes was calculated as described in [40], with *PbEF1α* as an internal reference gene. The primers are listed in Appendix A.

### 4.9. Molecular Cloning of PbCaM3/PbCML13

The specific primers for *PbCaM3*/*PbCML13* were designed and listed in Appendix A. High fidelity PCR amplification was performed using 2× TransStart FastPfu PCR SuperMix (AS221-01, Transgen, Beijing, China) with the reverse transcribed cDNA as a template, and then the fragment was recovered and purified.

### 4.10. Analysis of Drought Tolerance in PbCaM3/CML13 Transgenic Yeasts

Yeast ectopic expression analysis was performed as described in [52], with some modifications. Yeast INVSc1 receptor cells successfully transformed with recombinant and empty vector plasmids were inoculated into 1 mL of SC/-Ura liquid medium containing 2% *w/v* galactose to induce gene expression and incubated at 30 °C until OD = 0.3. Cultured yeast cells were serially diluted with ddH_2_O at a 10-fold gradient (1×, 10×, 100×, 1000×), and then 2 μL of the sample was spotted on SD/-Ura and SD/-Ura + 5 mM PEG solid medium. The yeast tolerance to drought stress was observed after 2 days of incubation at 30 °C.

### 4.11. Statistical Analysis

All data were statistically analyzed using SPSS software (v26.0). Significant differences in data related to the expression patterns of *PbCaM*s/*CML*s were determined by one-way ANOVA, and Duncan’s multiple comparisons test, as *p* < 0.05. Graphs were plotted using GraphPad Prism 9.3.

## 5. Conclusions

In this study, a total of five *PbCaM* and sixty *PbCML* genes were identified in the genome of *P. bournei*. Through further analyses of phylogeny, conserved motifs, gene structure, chromosomal location, gene duplication, promoter cis-acting elements, and expression characteristics in different tissues and abiotic stresses, it was found that the *PbCaM* genes were more conserved than the *PbCML* genes, and their functions were more general and diverse. The diverse expression of *PbCaM*/*CML* genes suggests that they may play important roles in the response of different tissues to various stresses. The overexpression of two stress-inducible genes, *PbCaM3* and *PbCML13*, significantly enhanced drought stress tolerance in yeast cells. The CaM/CML gene family in *P. bournei* was analyzed for the first time, which contributes to the understanding of the function of *PbCaM*s/*CML*s in regulating the response of plants to abiotic stresses, especially drought stress.

## Figures and Tables

**Figure 1 ijms-25-00545-f001:**
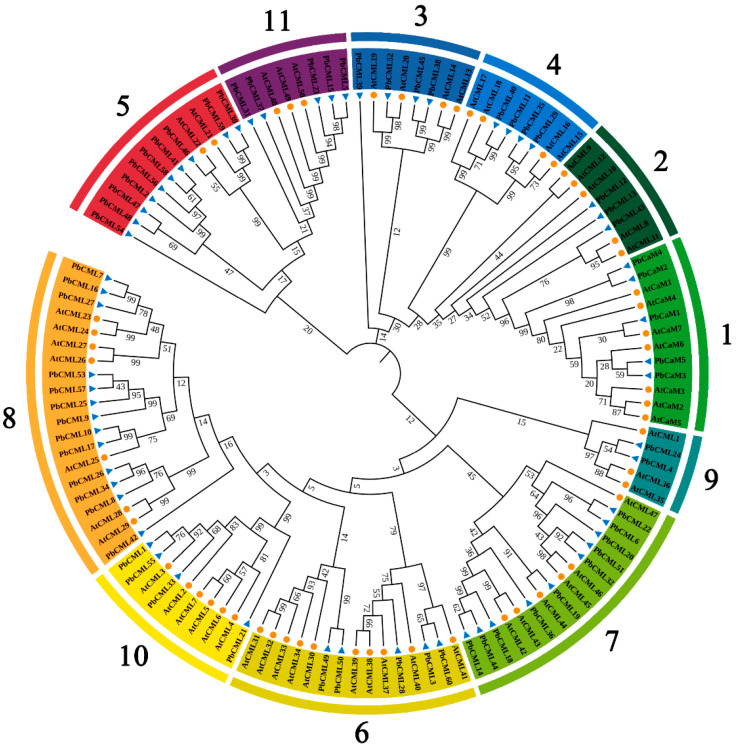
Phylogenetic tree of CaM/CML proteins in *A. thaliana* and *P. bournei*. Numbers 1–11 represent different subfamilies. Blue triangles represent PbCaM/CML proteins and orange circles indicate AtCaM/CML proteins. Different colored circles represent CaM/CML subfamilies.

**Figure 2 ijms-25-00545-f002:**
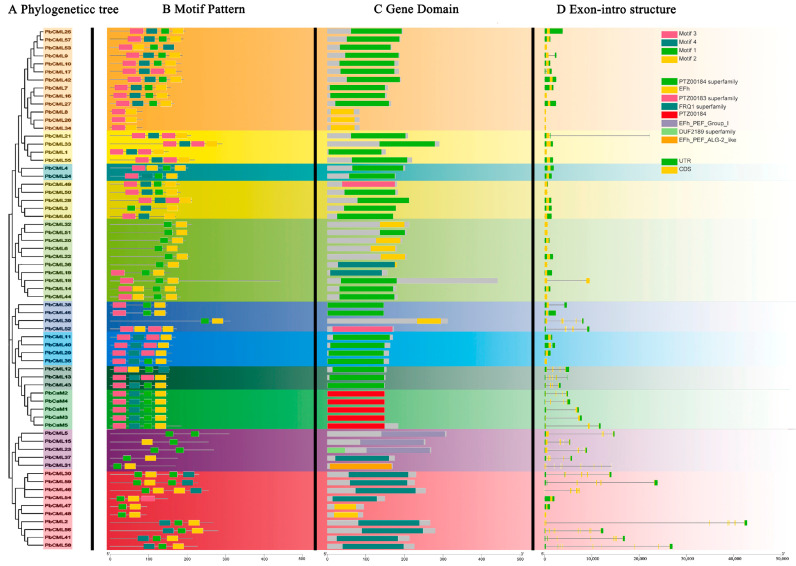
Phylogenetic tree, conserved motifs, gene domains, and gene structure of *PbCaM*s/*CML*s. (**A**): Phylogenetic tree of PbCaM/CML proteins. (**B**): Distribution of conserved motifs in PbCaM/CML proteins. (**C**): Types and distribution of gene domains in *PbCaM*s/*CML*s. (**D**): Gene structure of *PbCaM*s/*CML*s.

**Figure 3 ijms-25-00545-f003:**
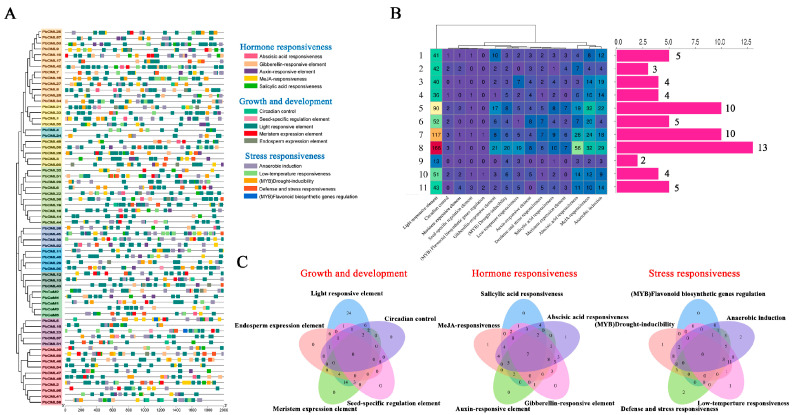
Predicted CREs of *PbCaM*s/*CML*s. (**A**): Distribution of CREs in the 2000 bp promoter region upstream of *PbCaM*s/*CML*s. Different colored squares represent different response elements. (**B**): The number of CREs in the promoter region of *PbCaM*s/*CML*s and their distribution in each subgroup. The horizontal coordinates of the heat map represent the different response elements, and the vertical coordinates represent 1–11 subfamilies. Warmer colors represent a greater number of response elements and cooler colors represent a smaller number of response elements. The horizontal coordinates of the bar graph represent 1–11 subfamilies and the vertical coordinates represent the number of *PbCaM*/*CML* genes. Numbers indicate the number of *PbCaM*s/*CML*s contained in each subfamily. (**C**): Venn diagram of CREs associated with hormone response, growth and development, and stress response. Numbers represent the number of *PbCaM*s/*CML*s containing the relevant response elements.

**Figure 4 ijms-25-00545-f004:**
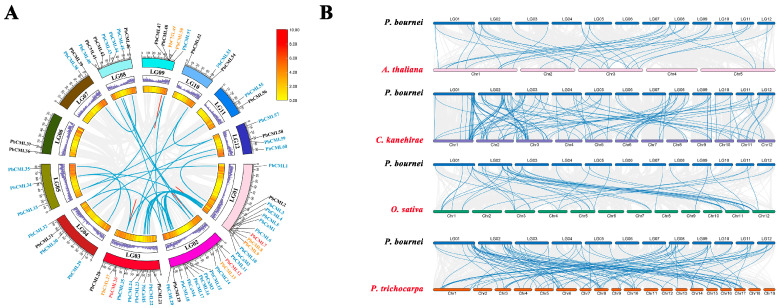
Homology analysis of *CaM*/*CML* genes within *P. bournei* and between *P. bournei* and *A. thaliana*, *O. sativa*, *C. kanehirae*, and *P. trichocarpa*. (**A**): Analysis of *CaM*s/*CML*s gene duplication in *P. bournei*. The outermost colored circle represents the 12 chromosomes (LG01-12) of *P. bournei*, and the inner 2 circles represent the gene density on each chromosome. The blue lines indicate the segmental replication gene pairs; the red lines indicate the tandem replication gene pairs. Blue-labeled *PbCaM*/*CML* genes are involved in segmental duplication, yellow-labeled genes are involved in tandem duplication, and red-labeled genes are involved in both segmental and tandem duplication. (**B**): Homologous *CaM*/*CML* genes between *P. bournei* and *A. thaliana*, *O. sativa*, *C. kanehirae*, and *P. trichocarpa*. The 12 chromosomes of *P. bournei*, 5 chromosomes of *A. thaliana*, 12 chromosomes of *O. sativa*, 12 chromosomes of *C. kanehirae*, and 19 chromosomes of *P. trichocarpa* are plotted with different colors. The blue lines represent *CaM*/*CML* gene pairs that are homologous among different species.

**Figure 5 ijms-25-00545-f005:**
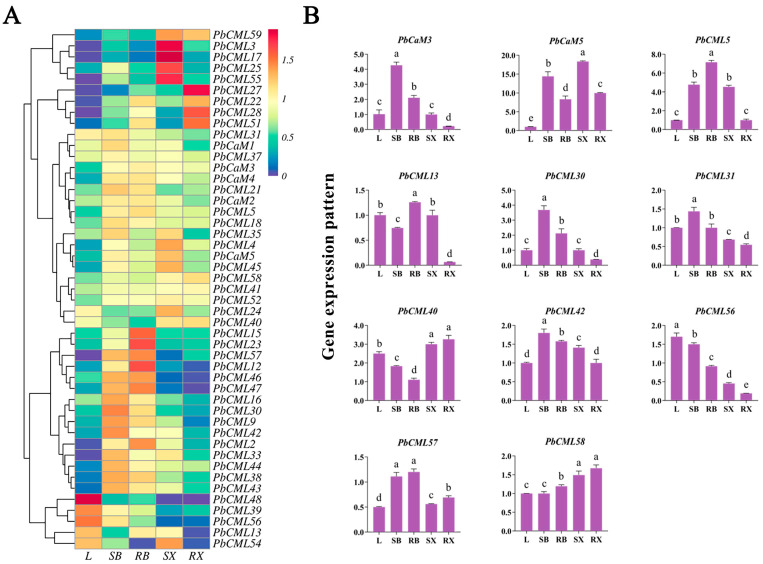
Expression patterns of *PbCaM*s/*CML*s in different tissues of *P. bournei* analyzed using transcriptome data and qRT-PCR. (**A**): Expression profiles of 11 *PbCaM*s/*CML*s in 5 different tissues (leaf, stem bark, root bark, stem xylem, and root xylem). (**B**): qRT-PCR validation of the expression patterns of 11 *PbCaM*s/*CML*s in 5 different tissues. Lowercase letters indicate significant differences, as *p* < 0.05.

**Figure 6 ijms-25-00545-f006:**
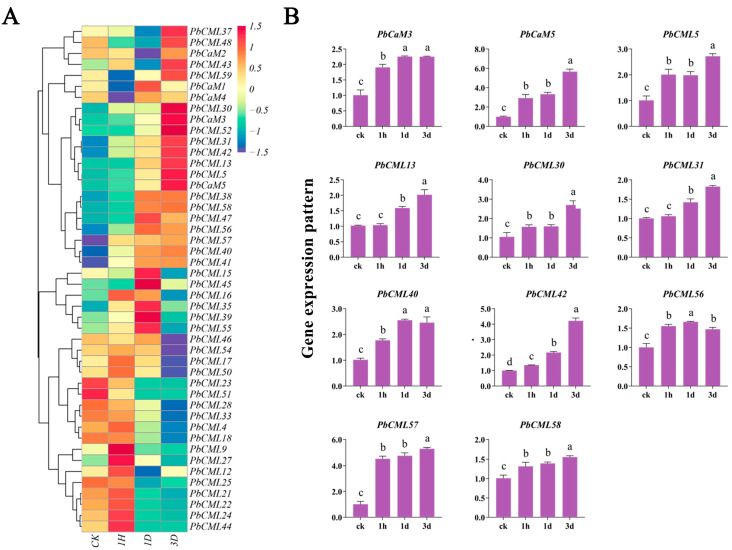
Expression patterns of *PbCaM*s/*CML*s under drought stress analyzed using transcriptome data and qRT-PCR. (**A**): Expression profiles of 11 *PbCaM*s/*CML*s under drought stress. (**B**): qRT-PCR validation of the expression patterns of 11 *PbCaM*s/*CML*s under drought stress. Lowercase letters indicate significant differences, as *p* < 0.05.

**Figure 7 ijms-25-00545-f007:**
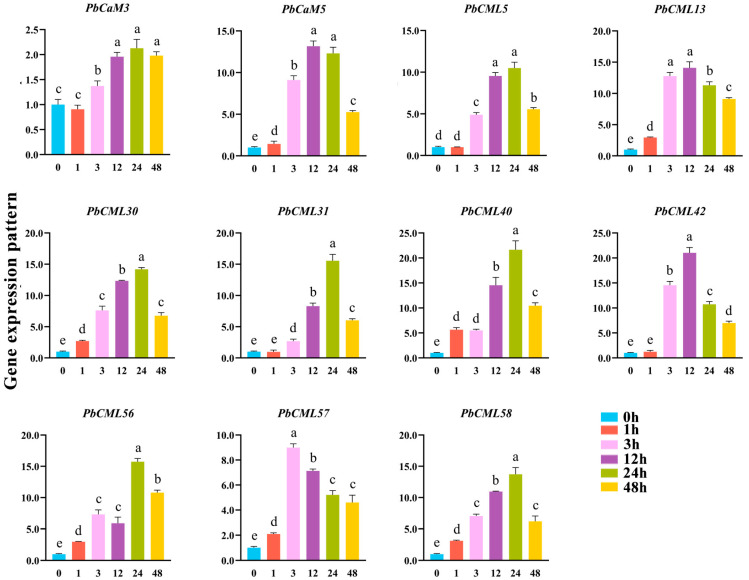
qRT-PCR validation of 11 *PbCaM*s/*CML*s under ABA treatment. Lowercase letters indicate significant differences, as *p* < 0.05.

**Figure 8 ijms-25-00545-f008:**
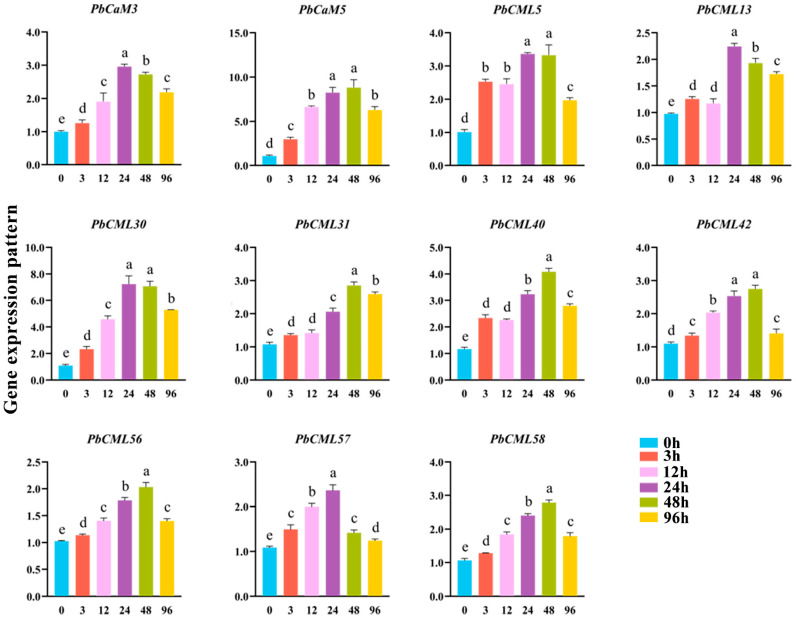
qRT-PCR validation of 11 *PbCaM*s/*CML*s under MeJA treatment. Lowercase letters indicate significant differences, as *p* < 0.05.

**Figure 9 ijms-25-00545-f009:**
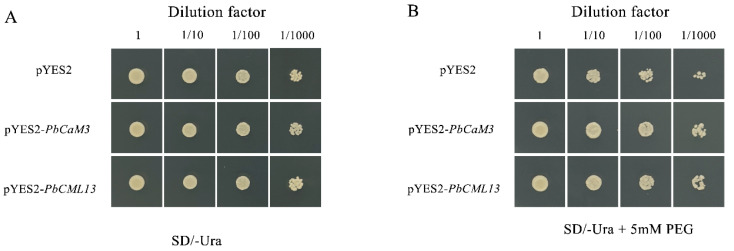
Analysis of drought stress tolerance of *PbCaM3*/*CML13* transgenic INVSc1 yeast. (**A**): Growth of yeast cells transformed with empty vector (PYES2) and PYES2-*PbCaM3*/*CML13* on SD/-Ura solid medium. (**B**): Growth of yeast cells transformed empty vector (PYES2) and PYES2-*PbCaM3*/*CML13* on SD/-Ura + 5 mM PEG solid medium.

## Data Availability

Data are contained within the article and Appendix A.

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
