# Peer review of "Genome-Wide Identification and Expression Analysis of Calmodulin and Calmodulin-like Genes, Revealing CaM3 and CML13 Participating in Drought Stress in Phoebe bournei"

_ijms, 2023, doi:10.3390/ijms25010545_

Round 1
Reviewer 1 Report
Comments and Suggestions for Authors
In the present manuscript, calmodulin and calmodulin-like gene families are well studied in in Phoebe bournei. The available bioinformatics tools were used to analyze structure and regulatory systems of CaM and CML family members. Also, functions of CaM3 and CML13 are well illustrated in response to drought stress. In my opinion, it has a good potential for publishing. Other comments:
- Gene family names should not be written in italics.
- INTRODUCTION is very well written.
- The resolution of figure 7 is low, please edit it. In addition, expression levels of controls can be removed from figure and they all can be merged and provided as 0h sample.
- Suggest adding it to line 329: “Partial duplications have probably affected the intron number in a gene. Besides, the number of intron/exon can alter the speed of the gene expression process and the mRNA processing, suggesting that genes with low number of intron/exons may be faster expressed (Arab et al, 2023; Yaghobi and Heidari, 2023).
- References: Arab et al, 2023: https://doi.org/10.3390/genes14030753
- Yaghobi and Heidari, 2023: https://doi.org/10.3390/genes14010202
- Line 391: provide the database that P. bournei proteins were extracted.
Author Response
Dear Editor and Reviewers
Thank you for handling our review process and for the reviewers' comments on our manuscript entitled " Genome-wide identification and expression analysis of calmodulin and calmodulin-like genes, revealing CaM3 and CML13 participating in drought stress in Phoebe bournei " (ijms-2784409). These comments were valuable and very helpful. We have carefully studied these comments and have made changes that we hope will meet with your approval. Based on the instructions you provided in your letter, we have uploaded the files of the revised manuscript. We have made minor revisions to the manuscript and the changes in the text are highlighted in yellow to improve the presentation at all levels of the manuscript. We have endeavoured to respond carefully to the reviewers' comments.
Sincerely
Junhong Zhang
Reviewer #1:
We are very grateful for your efforts in reviewing our paper and for the positive feedback you have given. Your summary of our work is accurate. We will respond to those issues and suggestions below.
Specific Comments
Q1. Gene family names should not be written in italics.
Response 1:Thank you for your suggestion. We have changed the italicization of gene family names in the revised version.
Q2. The resolution of figure 7 is low, please edit it. In addition, expression levels of controls can be removed from figure and they all can be merged and provided as 0h sample.
Response 2:Thank you for your suggestion. We have removed the expression levels of the controls and provided them as 0h samples in Figure 7 and increased the resolution of Figure 7 in the revised version.
Q3. Suggest adding it to line 329: “Partial duplications have probably affected the intron number in a gene. Besides, the number of intron/exon can alter the speed of the gene expression process and the mRNA processing, suggesting that genes with low number of intron/exons may be faster expressed (Arab et al, 2023; Yaghobi and Heidari, 2023).
Response 3:We deeply appreciate your suggestion. We have added “Partial duplications have probably affected the intron numbers in a gene. Besides, the number of introns/exons can alter the speed of the gene expression process and the mRNA processing, suggesting that genes with low number of introns/exons may be faster expressed” to line 329 in the revised version.
Q4. Line 391: provide the database that P. bournei proteins were extracted.
Response 4:Thank you for your suggestion. We have provided the results of PbCaM/CML proteins sequence extraction using BLAST and HMMER search tools in Supplementary file Table S8 in the revised version.
Reviewer 2 Report
Comments and Suggestions for Authors
English language is good.
Author Response
Dear Editor and Reviewers
Thank you for handling our review process and for the reviewers' comments on our manuscript entitled " Genome-wide identification and expression analysis of calmodulin and calmodulin-like genes, revealing CaM3 and CML13 participating in drought stress in Phoebe bournei " (ijms-2784409). These comments were valuable and very helpful. We have carefully studied these comments and have made changes that we hope will meet with your approval. Based on the instructions you provided in your letter, we have uploaded the files of the revised manuscript. We have made minor revisions to the manuscript and the changes in the text are highlighted in yellow to improve the presentation at all levels of the manuscript. We have endeavored to respond carefully to the reviewers' comments.
Sincerely
Junhong Zhang
Reviewer #2:
We appreciate your efforts and positive feedback in reviewing our manuscripts. In particular, the valuable comments on our work help us to better improve our manuscripts. We will respond to these comments and suggestions below.
Q1. Fig.2:
1). The font of legend probably needs to be increased.
2). Hard to see a phylogenetic tree.
3). Overlap of fig.1 and fig.2?
Response 1:Thank you for your valuable comments. The font of the legend in Figure 2 has been increased and the font of the phylogenetic tree has been bolded to make it look clearer in the revised version. Furthermore, part A in Figure 2 is duplicated from Figure 1, this is to better present the number and relationship of gene structure and conserved motifs within in the same subfamily and between different subfamilies.
Q2. Fig.3 hard to read. All the labels need to be increased. Fig.4 same problem as Fig.3
Response 2:Thank you for your valuable comments. Labels have been added to Figure 3 of the revised version to improve readability. In addition, we believe that the labeling of Figure 4 has been given sufficient description for a better reading of Figure 4.
Q3: Fig.5-8:
1). Font size is too small to read in panel ‘Gene expression pattern in fig.5, 6, 7, and 8’. Use combined figs instead of them.
2). Remove the consecutive lines in Fig.5B, 6B.
Response 3:Thank you for your valuable comments. We have enlarged the font and switched to a "Gene Expression Patterns" combination diagram in Figure 5-8, and removed the connecting lines in Figures 5B and 6B in the revised version.
Reviewer 3 Report
Comments and Suggestions for Authors
Dear Authors,
The submitted manuscript titled „Genome-wide identification and expression analysis of calmodulin and calmodulin-like genes, revealing CaM3 and CML13 participating in drought stress in Phoebe bournei” is generalny well-written. Nevertheless, I have found some imperfections, which (in my opinion) should be improved before an eventual publication. Please, find them below:
1. In my opinion the justification of choice of Phoebe bournei for investigations should be enlarged in the chapter Introduction.
2. I suggest to enlarge the description of a studied species. Such description should contain information about life form, lifespan, mode of reproduction, habitat affiliations and range of talon, as well as conservation status.
3. At the end of chapter Introduction the specific aims should be listed.
4. Figure 2-6 are illegible.
5. Please, look into the following publications. Perhaps, some of them would be help ful in manuscript improvments:
· Li et al. 2022. Leaf-transcriptome profiles of phoebe bournei provide insights into temporal drought stress responses. Front. Plant Sci., Sec. Plant Abiotic Stress, 13 https://doi.org/10.3389/fpls.2022.1010314
· Ge, Y., He, X., Wang, J. et al. Physiological and biochemical responses of Phoebe bournei seedlings to water stress and recovery. Acta Physiol Plant 36, 1241–1250 (2014). https://doi.org/10.1007/s11738-014-1502-3
· Ma et al. 2023. Genome Identification and Evolutionary Analysis of LBD Genes and Response to Environmental Factors in Phoebe bournei. nt. J. Mol. Sci. 2023, 24, 12581. https://doi.org/10.3390/ijms241612581
· Liao, W., Tang, X., Li, J. et al. Genome wide investigation of Hsf gene family in Phoebe bournei: identification, evolution, and expression after abiotic stresses. J. For. Res. 35, 11 (2024). https://doi.org/10.1007/s11676-023-01661-y
Author Response
Dear Editor and Reviewers
Thank you for handling our review process and for the reviewers' comments on our manuscript entitled " Genome-wide identification and expression analysis of calmodulin and calmodulin-like genes, revealing CaM3 and CML13 participating in drought stress in Phoebe bournei " (ijms-2784409). These comments were valuable and very helpful. We have carefully studied these comments and have made changes that we hope will meet with your approval. Based on the instructions you provided in your letter, we have uploaded the files of the revised manuscript. We have made minor revisions to the manuscript and the changes in the text are highlighted in yellow to improve the presentation at all levels of the manuscript. We have endeavored to respond carefully to the reviewers' comments.
Sincerely
Junhong Zhang
Reviewer #3:
We appreciate your efforts and positive feedback in reviewing our manuscripts. Comments on our work are valuable and accurate and help us to improve our manuscripts. We will respond to your questions and suggestions below.
Q1. In my opinion the justification of choice of Phoebe bournei for investigations should be enlarged in the chapter Introduction.
Response 1:Thank you for your valuable advice. We have added the reasons for choosing P. bournei for the survey to the 'Introduction' chapter of the revised manuscript: Its wood is known as "noble wood" because of its strong resistance to decay, special fragrance and unique golden-yellow texture, and it is also an important garden ornamental and landscaping tree species. P. bournei is a widespread afforestation species in the mountainous regions of southern China. Therefore, the study of the response mechanism of P. bournei to drought stress is very important for the cultivation and conservation of this species.
Q2. I suggest to enlarge the description of a studied species. Such description should contain information about life form, lifespan, mode of reproduction, habitat affiliations and range of talon, as well as conservation status.
Response 2:We deeply appreciate your suggestion. We have added a description of P. bournei in the last paragraph of the introduction to the revised manuscript: Phoebe bournei is a subtropical evergreen broad-leaved tree species, preferring humidity and shade, with a long lifespan and slow growth, and seed reproduction is the main way of its natural renewal. P. bournei is mainly found in areas with a warm and humid climate and abundant rainfall, such as Jiangxi, Fujian, Zhejiang, Guangdong, Guangxi, Guizhou, Hubei and Hunan provinces. However, in recent years, over-harvesting and habitat degradation have led to a reduction in the size and fragmentation of the natural population of P. bournei, seriously threatening its survival. At present, a number of protected areas have been established within its range and P. bournei is being propagated and promoted as a silvicultural species.
Q3. At the end of chapter Introduction the specific aims should be listed.
Response 3:Thank you for your valuable comments. We have added a characterization of the protein physicochemical properties, phylogenetic relationships, gene structure and conserved motifs, chromosomal localization and homology, and cis-acting elements in the promoter regions of the PbCaM/CML gene family at the end of the "Introduction" chapter of the revised manuscript.
Q4. Figure 2-6 are illegible.
Response 4:Thank you for your suggestion. In the revised version, the resolution of Figure 2-6 has been increased for better readability.